# Promoting multiexciton interactions in singlet fission and triplet fusion upconversion dendrimers

Guiying He [1,2], Emily M. Churchill [3], Kaia R. Parenti[3], Jocelyn Zhang[3], Pournima Narayanan [4,5], Faridah Namata [6], Michael Malkoch [6], Daniel N. Congreve [4], Angelo Cacciuto[3], Matthew Y. Sfeir [1,2] ✉ & Luis M. Campos [3] ✉

Singlet fission and triplet-triplet annihilation upconversion are two multi-exciton processes intimately related to the dynamic interaction between one high-lying energy singlet and two low-lying energy triplet excitons. Here, we introduce a series of dendritic macromolecules that serve as platform to study the effect of interchromophore interactions on the dynamics of multiexciton generation and decay as a function of dendrimer generation. The dendrimers (generations 1–4) consist of trimethylolpropane core and 2,2-bis(methylol) propionic acid (bis-MPA) dendrons that provide exponential growth of the branches, leading to a corona decorated with pentacenes for SF or anthracenes for TTA-UC. The findings reveal a trend where a few highly ordered sites emerge as the dendrimer generation grows, dominating the multiexciton dynamics, as deduced from optical spectra, and transient absorption spectroscopy. While the dendritic structures enhance TTA-UC at low annihilator concentrations in the largest dendrimers, the paired chromophore interactions induce a broadened and red-shifted excimer emission. In SF dendrimers of higher generations, the triplet dynamics become increasingly dominated by pairwise sites exhibiting strong coupling (Type II), which can be readily distinguished from sites with weaker coupling (Type I) by their spectral dynamics and decay kinetics.

Recent developments in organic optoelectronic materials have led to a growing interest in understanding how multiexcitonic states impact devices and photochemical transformations[1–4]. Unlike traditional photochemical processes, multiexciton systems inherently involve two or more organic chromophores that can undergo two types of biexcitonic mechanisms[5]. When the energy of biexciton is less than that of the lowest singlet state, the chromophores exhibit singlet fission (SF) where a photoexcited singlet exciton ($S_1$) decays via a spin and energy conserved process to a coupled triplet pair ($[T_1T_1]$) delocalized across two or more chromophores[6,7]. The $[T_1T_1]$ state can then be harvested directly or after dephasing into two uncorrelated free triplets ($2 \times T_1$)[8–10]. This process is the basis for proposed solar cell architectures that utilize this carrier multiplication process to drive higher power conversion efficiencies[1,4,11–13]. On the other hand, triplet–triplet annihilation upconversion (TTA-UC) represents the time-reversed process of SF[14]. Also known as triplet fusion, TTA-UC occurs in chromophores for which

[1]Department of Physics, Graduate Center, City University of New York, New York, NY 10016, USA. [2]Photonics Initiative, Advanced Science Research Center, City University of New York, New York, NY 10031, USA. [3]Department of Chemistry, Columbia University, New York, NY 10027, USA. [4]Department of Electrical Engineering, Stanford University, Stanford, CA 94305, USA. [5]Department of Chemistry, Stanford University, Stanford, CA 94305, USA. [6]KTH Royal Institute of Technology, Department of Fibre and Polymer Technology, SE-100 44 Stockholm, Sweden. ✉e-mail: msfeir@gc.cuny.edu; lcampos@columbia.edu

the biexciton $[T_1T_1]$ generated by the encounter of two free triplets is slightly energetically higher than the bright $S_1$ state by a spin-allowed transition. When combined with an appropriate triplet sensitizer, these systems can exhibit large apparent anti-Stokes shifts, effectively converting two low-energy photons into a single higher energy one[15–18]. While this process has been invoked to harvest photons with energy lower than the bandgap of a photovoltaic device, it has also been found to initiate photochemical transformations that are useful in a variety of applications.

In biexcitonic systems, the net spin, energy, and formation/decay dynamics depend sensitively on the intricacies of chromophore-chromophore coupling interactions[19–24]. Notably, the coupling can be modulated using through-bond interactions by covalently linking two or more chromophores or through-space by adjusting interchromophore interactions to achieve wave-function overlap. With this level of control, a wide variety of intramolecular SF (iSF) dimers, oligomers, and polymers have been developed to demonstrate how through-bond covalent interactions can be optimized through molecular engineering[9,25–30]. In these systems, structure–property relationships have emerged that describe the dependence of $[T_1T_1]$ formation and decay dynamics on proximity, relative geometry, and connectivity of the chromophores. However, it has been a challenge to control multi-chromophore interactions and understand intermolecular SF (xSF) in condensed phase systems[21,23,31,32]. In xSF, through-space interactions dominate, and the upconversion and downconversion dynamics of the $[T_1T_1]$ state depend on the morphology, crystal structure, and grain size. However, small molecule dimers and polymers with pendent motifs can be enabled either through-space SF or TTA-UC[33–39]. Important differences in the multiexciton dynamics of through-bond and through-space systems have been identified in these studies. Most importantly, a greater diversity of interfacial electronic states arises in through-space systems, including excimer and charge-transfer states, that can be parasitic for the desired multiexciton conversion processes. These parasitic effects include a reduction of the apparent anti-Stokes shift in TTA-UC and a reduction of the SF yield[19,40–43].

To date, molecular engineering approaches to control through-space interactions have been generally limited to linear multichromophores, star molecules, or one-dimensional arrangements of chromophores in polymers[33,34,44]. However, molecular crystals have three-dimensional wavefunctions for the singlet and triplet pair states. Interestingly, pendent polymer architectures exhibit structural degrees of freedom that cause significant heterogeneity in chromophore alignment, as opposed to the rigid environment of crystals, glasses, and even covalent frameworks[34,45]. The heterogeneity is reflected in the multiexciton dynamics and impacts the overall rates of multiexciton formation and decay. It is also worth noting that interchromophore coupling strengths determine the speed of triplet pair and triplet diffusion processes. These transport processes are difficult to study in supramolecular systems and non-existent in dimers. The shutdown of diffusion–reliant pathways is especially problematic for TTA-UC systems, where the efficiency suffers at low annihilator concentrations. To address these challenges, we posit that macromolecular engineering approaches can be extended to encompass three-dimensional covalent chromophore assemblies in which the multiexciton dynamics can be modulated with varying numbers of chromophores and 3D arrangements. In particular, we focus on dendritic architectures, which provide exquisite control of structure, size, and end-group functionality. The precise degree of branching in dendrimers varies by the identity of the core and growth generation ([Gn], where n represents the generation number)[46,47]. Yet nothing is known about multiexciton processes in dendrimers. The ability to tailor the structure, shape, and surface functional groups renders these macromolecules important systems in catalysis, drug delivery, MRI contrast agents, and porogens, in addition to a wealth of other studies[48].

Here, we demonstrate that dendrimers create a structurally ordered and well-defined non-linear macromolecular 3D architecture that serves as an effective framework for investigating fundamental multiexciton dynamics in multichromophore systems. We designed two different classes of polyester dendrimers consisting of a weakly coupled extended network of pairwise chromophore units per dendron in the corona that permits both rapid multiexciton formation and efficient diffusion. In one class of acene-based dendrimers [Gn]-$Ac_x$ (Fig. 1), the corona consists of 6,13-bis(triisopropylsilylethynyl)-pentacene (Ac = TIPS-pentacene), a prototypical SF chromophore. In another class, the corona consists of 9,10-bis(triisopropylsilylethynyl)-2-anthracene (Ac = TIPS-anthracene), a prototypical TTA-UC chromophore. Taken together, the 3D spatial arrangement of the acenes can mimic the molecular packing arrangements of a multichromophore segment of condensed phase systems. However, systematically increasing the number of chromophores through its generation provides a key advantage in allowing detailed spectroscopic studies that are sensitive to the coupling strength of neighboring chromophores. Here we find a systematic change in the multiexciton dynamics as a function of generation due to the emergence of a small number of hot spots in the branching units of the corona that dominate the pairwise multiexciton dynamics. This suggests efficient exciton transport within the three-dimensional network to well-defined sites, a process that can sufficiently mitigate the negative effects of heterogeneities in other dynamic systems.

## Results and discussion

We synthesized four generations of polyester dendrimers containing TIPS-pentacene or TIPS-anthracene units in the periphery (Fig. 1, [Gn]-$Ac_x$, where the generation number/acene units are given by $n/x = 1/6, 2/12, 3/24,$ and $4/48$). Each dendrimer consists of a trimethylolpropane (TMP) core and 2,2-bis(methylol)propionic acid (bis-MPA) dendrons that provide flexibility to the branches. Each hydroxyl-functionalized bis-MPA dendrimer was modified with 4-bromobenzoic acid via fluoride-promoted esterification chemistry (FPE)[49–51] and subsequently subjected to a Suzuki–Miyaura cross-coupling with a boronic pinacol ester (BPin)-derivatized pentacene or anthracene to furnish the desired acene-functional dendrimers (details in Supplementary Figs. 27–29). The [Gn]-$Ac_x$ dendrimers were purified by silica gel column chromatography. The degree of functionalization ranged from 90% to quantitative, where the lowest values were obtained with [G4]-$Ac_{48}$ due to the large number of peripheral functional groups. The smallest dendrimer, [G1]-$Ac_6$, was quantitatively functionalized, and all dendrimers were characterized by nuclear magnetic resonance (NMR) and matrix-assisted laser desorption/ionization-time of flight (MALDI-TOF) mass spectrometry. The dispersity, Đ, was obtained by size exclusion chromatography (SEC), yielding values ranging from 1.01 to 1.05, which are representative of well-defined dendrimers.

Numerical simulations suggest that the equilibrium conformational properties of the dendrimers permit through-space chromophore-chromophore interactions that are favorable for multiexciton formation. We used molecular dynamics simulations[52] of a simple spring-and-ball mesoscopic model discussed in detail in the Supplementary Discussion. The numerical results suggest that, overall, the pentacenes are pointing away from the center of the dendrimer. However, fluctuations away from the radial direction are larger in the [G1]-$P_6$ and [G2]-$P_{12}$ and more moderate in [G3]-$P_{24}$ and [G4]-$P_{48}$. These trends arise from the weaker binding energy and reduce steric aligning of the pentacenes at low generations. The angular probability distributions of the dendrimer models, as well as the radial positional probability distribution of the pentacene's group center of mass, are plotted in Supplementary Figs. 9–11. With the increase in generation number and density of chromophores, some of the pentacenes also recede from the periphery and relocate closer to the dendrimer core.

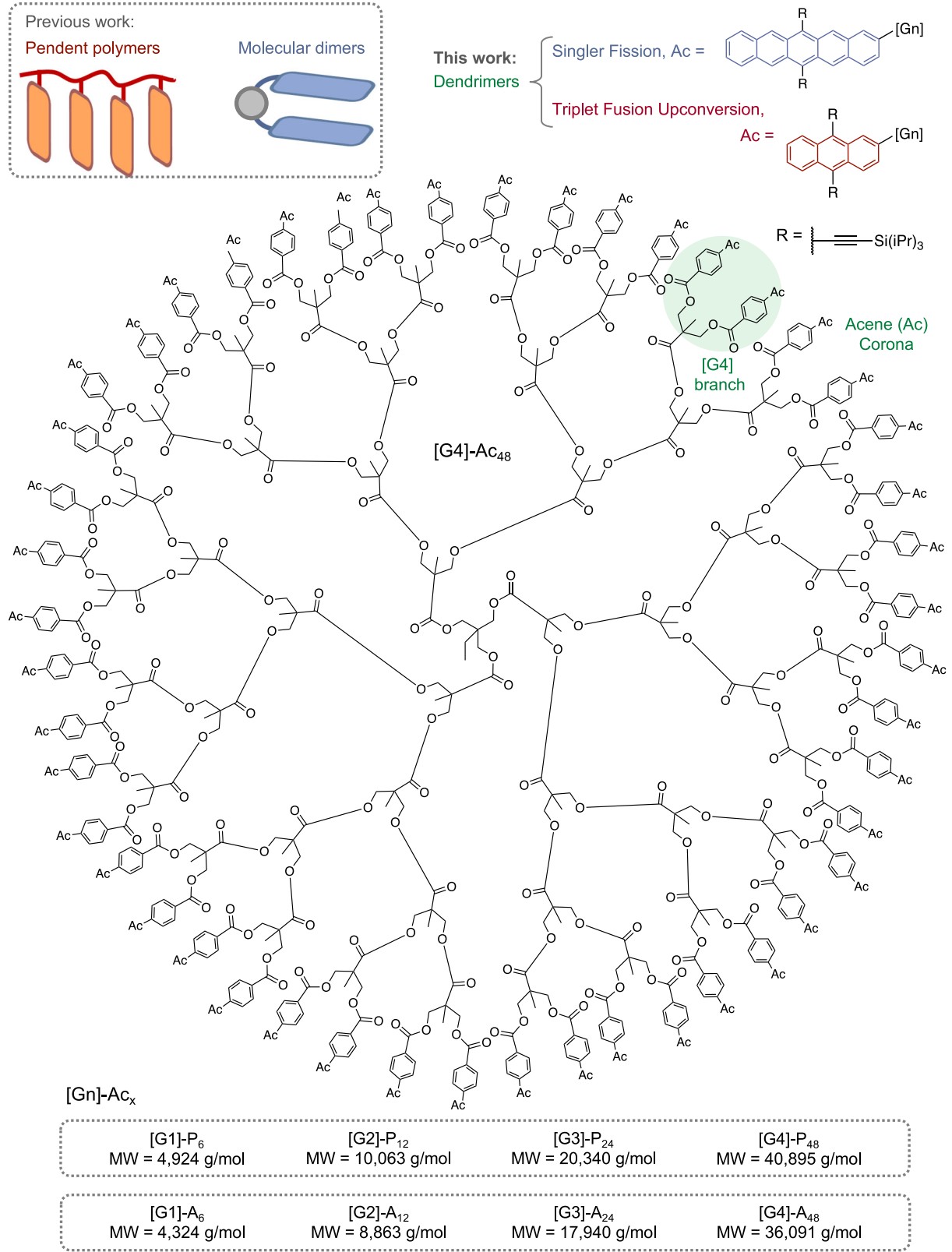

**Fig. 1 | Molecular and macromolecular architectures that enable through-space interactions in multiexciton systems with two or more chromophores.** Chemical structure of [G4]-Ac$_{48}$, where Ac represents either the acenes TIPS-pentacene for singlet fission-active dendrimers or TIPS-anthracene for triplet−triplet annihilation upconversion-active dendrimers. The molecular mass of each of the four generations of pentacene and anthracene dendrimers studied here is listed for comparison.

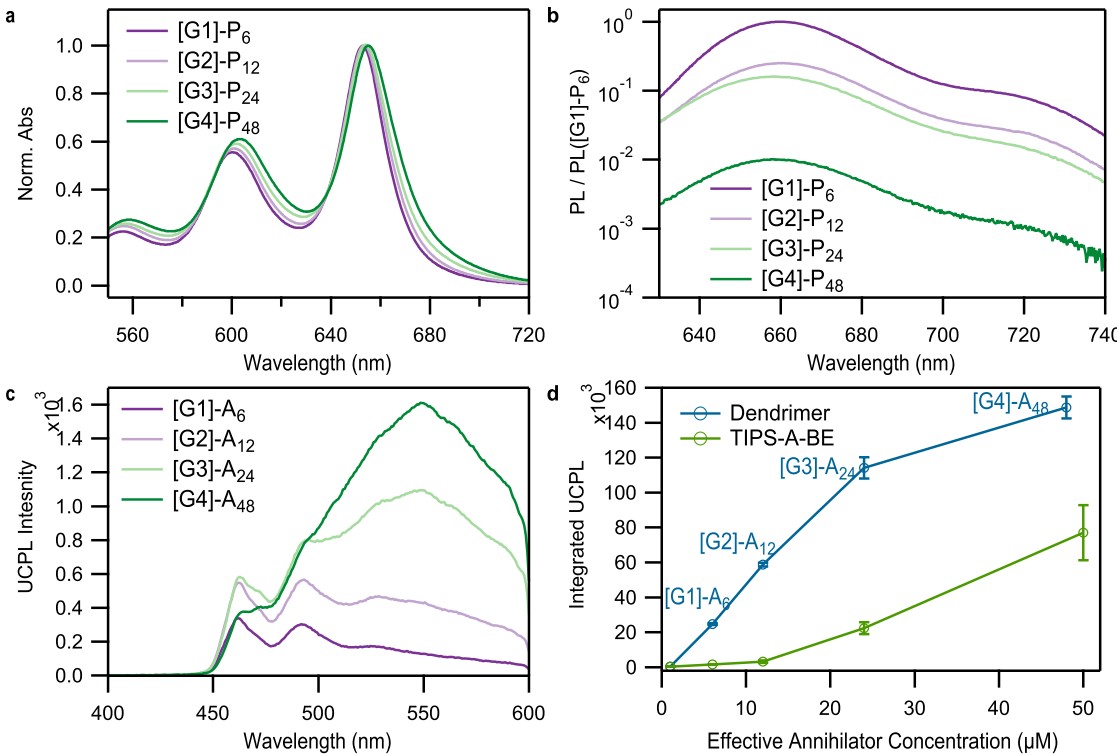

**Fig. 2 | Steady-state spectra of the dendrimers. a** Normalized absorption of the SF dendrimers [G1]-P$_6$, [G2]-P$_{12}$, [G3]-P$_{24}$, and [G4]-P$_{48}$ in the region corresponding to the $S_0 \rightarrow S_1$ transition. As the generation becomes larger, we observe a small red-shift of the peak maxima, enhanced NIR absorption, and an increase in the ratio between the 0–1 vibronic peak near 600 nm and the 0–0 vibronic peak near 654 nm. (**b**) The relative photoluminescence quantum yield (PLQY) decreases for larger generations, to 0.25 in [G2]-P$_{12}$, 0.16 in [G3]-P$_{24}$, and 0.01 in [G4]-P$_{48}$ compared to [G1]-P$_6$, consistent with the emergence of rapid singlet fission. The

excitation wavelength used here was 590 nm. **c** Upconversion photoluminescence spectra of anthracene dendrimers measured at 1 μM with 50 μM PdTPTBP sensitizer in degassed chloroform with excitation at 635 nm and **d** values of UCPL intensity spectra integrated from 400 to 600 nm for anthracene dendrimers measured at 1 μM and analogous anthracene monomer, plotted against relative annihilator concentration. The error bars represent the standard deviations of the measured values ($n = 3$).

The inward fluctuations of the pentacenes release some of the stresses building up at the dense perimeter, thus losing some of the steric alignment provided by neighboring outer branches (Supplementary Fig. 11). Though the chemical structure of the dendrimer ensures pairwise chromophore interactions through the short propionic acid linker, our calculations suggest that conformational relaxation in higher generations may permit the formation of larger clusters. As a result, we expect rich photophysical behavior to emerge in these dynamic, well-defined macromolecular systems.

In agreement with numerical simulations, we find that the linear absorption spectra suggest stronger interchromophore coupling in higher generations (Fig. 2a). Within the pentacene dendrimer series, the maxima of the $S_0 \rightarrow S_1$ absorption undergoes a small red-shift, from 653 nm in [G1]-P$_6$, to 655 nm in [G4]-P$_{48}$. More noticeable is the broadening of the transition as the generation increases, leading to an increase in the absorption strength of the tail. The enhanced tail absorption strength is consistent with greater $\pi$-stacking interactions that delocalize the singlet exciton. Following a procedure previously used to quantify chromophore-chromophore coupling strengths in pentacene nanoparticles[53,54], we compare the absorption coefficient in the spectral region corresponding to monomer-like absorption at ~655 nm to the region corresponding to crystalline-like domains at ~703 nm: $\varepsilon_{703}/\varepsilon_{655}$. This value ranges from $2.5 \times 10^{-2}$ in [G1]-P$_6$ to $6.7 \times 10^{-2}$ in [G4]-P$_{48}$, indicating that a high percentage of ordered domains are present in higher generations. Additional evidence for enhanced interchromophore coupling comes from the increased ratio of 0–1 to 0–0 vibronic peaks of the $S_1$ exciton in higher generations, consistent with the formation of H-type aggregates[55,56]. The ratio

$I_{0–1}/I_{0–0}$ increases from 0.5 in monomeric TIPS-pentacene to 0.56 in [G1]-P$_6$ to 0.62 in [G4]-P$_{48}$. We observe similar trends for anthracene-based dendrimers that suggest stronger interchromophore interactions as the number of acene units increases (Supplementary Fig. 1). As the generation increases, the lowest-energy exciton absorption broadens, exhibits a weak red-shift, and decreases in intensity relative to the first vibronic overtone.

While the exciton absorption is weakly impacted by enhanced ordering in higher-generation dendrimers, the multiexciton dynamics are highly dependent on small changes in the local ordering. In pentacene dendrimers, an increase in the fraction of crystalline-like domains with higher generation should strongly quench the time-integrated photoluminescence due to the emergence of a competing (non-radiative) SF process that generates a triplet pair multiexciton. Indeed, we observe both subtle spectral broadening on the NIR side of the emission peak (Supplementary Fig. 1) and large changes in the overall photoluminescence quantum yields (PLQY) consistent with larger aggregates in higher generations. We find that after normalizing the PL data to its respective absorbance value at 590 nm, the relative emission intensity (PL/PL([G1]-P$_6$)) is reduced to 0.25 in [G2]-P$_{12}$, 0.16 in [G3]-P$_{24}$, and 0.01 in [G4]-P$_{48}$, as compared to [G1]-P$_6$ (Fig. 2b). This quenching is consistent with the emergence of rapid SF in pentacene aggregates. From these results, we conclude that inter-chromophore coupling increases with the higher generations of dendrimers, suggesting a more favorable alignment for facilitating SF.

In anthracene dendrimers, the energy level ordering favors triplet–triplet annihilation upconversion over SF. We find that this process is similarly sensitive to subtle changes in interchromophore

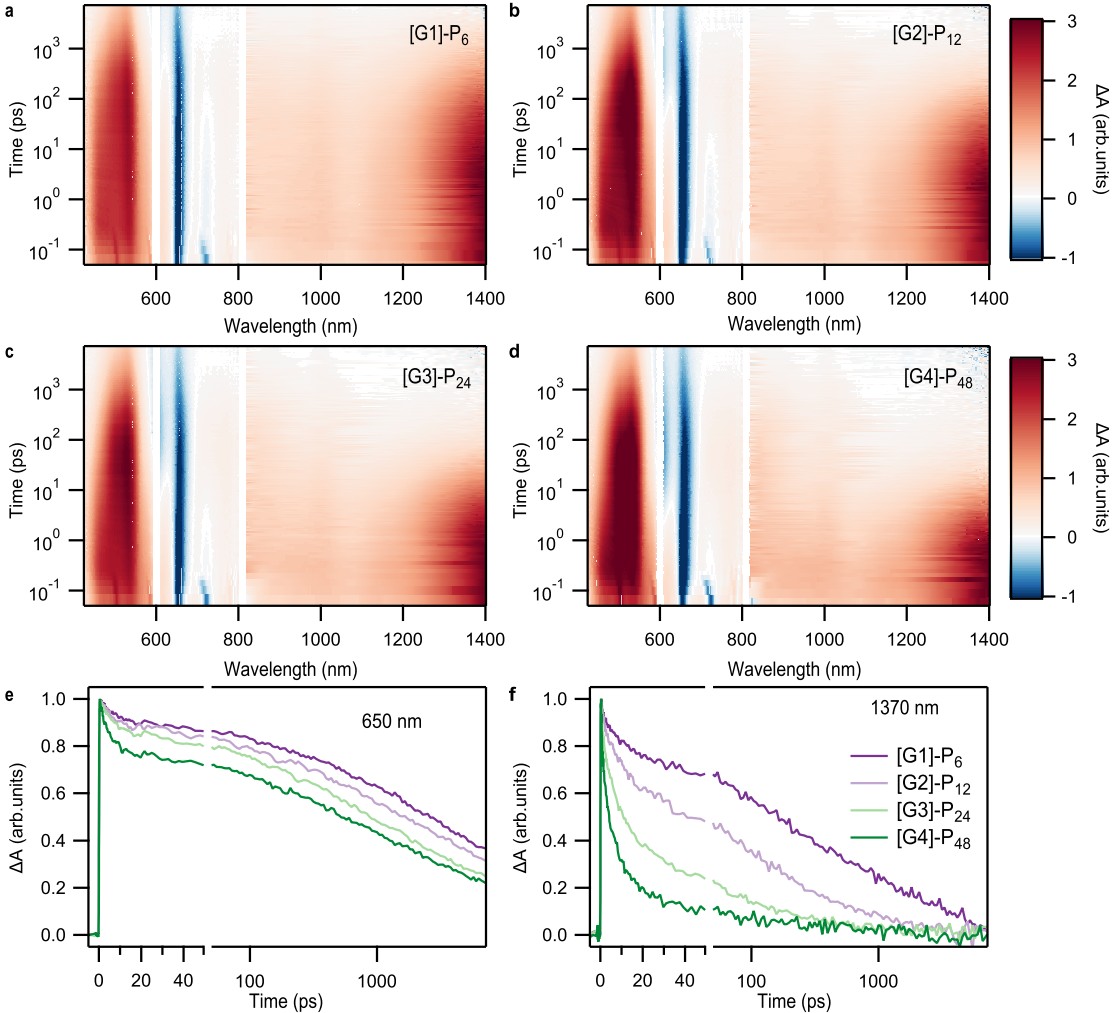

**Fig. 3 | Femtosecond transient absorption spectra and kinetics of pentacene dendrimers.** Raw femtosecond transient absorption data in the visible and NIR region of **a** [G1]-P$_6$, **b** [G2]-P$_{12}$, **c** [G3]-P$_{24}$, and **d** [G4]-P$_{48}$. Normalized kinetics show that the **e** decay of the ground state bleach feature near 650 nm is independent of the **f** singlet exciton decay that can be used to quantify the SF time constants.

coupling. Unlike pentacene, the PLQY of the anthracene exciton is minimally affected by the generation of the dendrimer since no strong competing non-radiative pathways emerge (Supplementary Table 1). However, when paired with an appropriate triplet sensitizer, the dendrimers serve as multi-annihilator macromolecules that facilitate TTA-UC to an emissive singlet state in a dilute solution. As such, the anthracene dendrimers provide a platform to study TTA-UC in a series of compounds that contain discrete numbers of annihilators covering a range of local annihilator concentrations and interchromophore coupling. For evaluating upconverted photoluminescence (UCPL) in dendrimers, we measured solutions of 1 μM dendrimer mixed with 50 μM palladium (II) tetraphenyltetrabenzoporphyrin (PdTPTBP) sensitizer, such that the local annihilator concentration depends on the dendrimer generation. To quantify enhancements provided by the macromolecular architecture, we also measured UCPL for mixed anthracene monomer/sensitizer solutions at comparable concentrations to each generation of the dendrimers (Fig. 2c).

As in pentacene, subtle changes of the linear absorption spectra with increased dendrimer generation are accompanied by significant changes in the UCPL spectra and overall quantum yield that reflect subtle changes in the multiexciton dynamics. As the generation increases, the upconverted emission evolves from a monomer-like emission spectrum with its maximum near 462 nm to an excimer-like

state emitting broadly between 500 and 600 nm (Fig. 2c). The excimer emission results from face-on interactions between two or more anthracene monomers and reflect increased order in the emissive sites within the dendrimer[40]. In [G4]-A$_{48}$, the monomer-like emission is barely discernable such that highly ordered sites are solely responsible for driving UCPL processes. While excimers are largely assumed to be parasitic to the UCPL process, we find that they are only weakly quenching. In the low concentration limit, the relative qualitative TTA-UC efficiency, as given by the integrated UCPL spectra between 400 and 600 nm, increases with increased dendrimer generation (Fig. 2d). The TTA-UC efficiency improves rapidly from [G1]-A$_6$ to [G3]-A$_{24}$ but starts to saturate in [G4]-A$_{48}$, indicating an overall decrease in the TTA-UC efficiency per effective annihilator concentration. This is coincident with the fact that excimers completely dominate the emission spectrum of [G4]-A$_{48}$. The effect of excimer formation on the efficiency normalized by effective annihilator concentration can be readily determined by comparing low and high generations (Fig. 2d). We find that the yield of excimer emission in [G4]-A$_{48}$ is about 40% smaller than that of monomer-like sites in [G2]-A$_{12}$. For applications, the detrimental effect of excimer formation may come from the reduction of the apparent anti-Stokes shift rather than the overall yield. Notably, at the very low annihilator concentrations used in this study, all dendrimers produced a larger UCPL signal than the monomer due to the

**Table 1 | Time constants of the triplet formation and decay**

| Dendrimer | TT^II | | TT^I | | T₁ |
|---|---|---|---|---|---|
| | Formation (ps) | Decay (ps) | Formation (ps) | Decay (ns) | Decay (µs) |
| [G1]-P₆ | 8.6 | 970 | 101.1 | 254 | 27.6 |
| [G2]-P₁₂ | 7.1 | 638 | 72.1 | 298 | 26.2 |
| [G3]-P₂₄ | 5.2 | 530 | 35.0 | 208 | 21.4 |
| [G4]-P₄₈ | 2.4 | 452 | 13.4 | 90 | 21.3 |

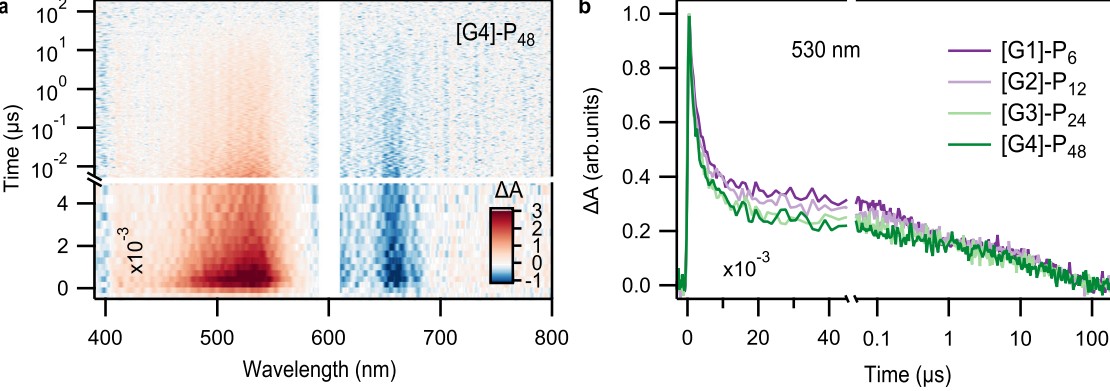

**Fig. 4 | Nanosecond transient absorption spectroscopy of pentacene dendrimers. a** Transient absorption data of [G4]-P₄₈ spanning nanosecond to microsecond time scales along with **b** kinetic cuts at 530 nm near the peak of the triplet excited state absorption feature. These data show a long-lived triplet pair and triplet exciton population whose average behavior can be characterized using two first-order time constants spanning 100–250 ns and 21–27 µs. In both cases, the time constants become shorter for higher generations.

lower probability of productive intermolecular collisions of the annihilator molecules. At high concentrations, the yields of [G2]-A₁₂ are similar to the monomer (Supplementary Fig. 4). These results suggest that dendritic systems can be designed that optimize interchromophore coupling to enhance TTA-UC yields at low concertation while suppressing excimer dynamics.

While the above data clearly show that there are changes in the degree of aggregation as a function of generation, additional insight into the spatial distribution and heterogeneity of interchromophore interactions can be determined using time-resolved optical measurements. Here, we focus on the pentacene dendrimers to quantify the emergence of multiexciton photophysics via their SF dynamics. We find that each of the four different dendrimer generations undergo SF, with the overall rate increasing in later generations. The femtosecond transient absorption spectra from the visible to near-infrared region of four dendrimers (Fig. 3a–d) exhibit characteristic spectral signatures for the singlet and triplet exciton that can be used to track the exciton conversion process. Immediately after excitation, the transient absorption spectra of dendrimers are associated with the population of the singlet exciton. This is characterized by a ground state bleach signal (GSB) around 650 nm and a broad excited state absorption (ESA) in the blue (430–580 nm) and NIR region (~1400 nm). In addition, we observed a weak stimulated emission signal (SE) around 725 nm. As the singlet decays, we observe a corresponding rise of the triplet ESA, with a strong peak around 530 nm and a weaker peak near 1000 nm. In these SF dendrimers, the triplet ESA is broad and exhibits significant overlaps with the singlet ESA in this region. As a result, the NIR ESA is a more direct marker than the visible triplet ESA for the decay of the singlet population, as it exhibits minimal overlap with triplet transitions. In all generations, fast singlet state decays without significant loss of the GSB signal, and triplet–triplet annihilation process is observed, indicating that SF is occurring.

SF becomes significantly faster as the generation becomes larger, as can be readily seen in the transient absorption kinetics near 1370 nm (Fig. 3f). Unsurprisingly, the triplet formation dynamics for all generations suggest that the magnitude of chromophore–chromophore interactions are not uniform, leading to multiexponential singlet decays. On average, we can capture the average SF kinetics with two characteristic time constants, which both decrease monotonically with increasing generation size. A fast time constant of 8.6 ps in [G1]-P₆, 7.1 ps in [G2]-P₁₂, 5.2 ps in [G3]-P₂₄, and 2.4 ps in [G4]-P₄₈ roughly corresponds to SF in relatively well-ordered regions of the dendrimer. A slower time constant of 101 ps, 72 ps, 35 ps, and 13 ps, [G1]-P₆-[G4]-P₄₈, respectively, corresponds to regions for which interchromophore coupling is weaker. As discussed above, these changes in rates are accompanied by large changes in the PLQY. This data implies that in earlier generations, particularly [G1]-P₆ and [G2]-P₁₂, a much larger fraction of singlet excitons is generated in regions with weak interchromophore coupling where radiative emission is competitive with SF. However, the low PLQY in [G4]-P₄₈ indicates that nearly all photoexcited singlets undergo rapid SF in higher generations.

The local chromophore packing heterogeneity suggested by the SF dynamics is also apparent in the decay of the triplet pair. For example, an extremely efficient triplet pair population decay process is apparent in the normalized GSB decay curves (Fig. 3e) for all generations on time scales of ~1 ns. These decays exhibits a weaker generation dependence than other excited state processes in these compounds (Table 1). The generation-independent kinetics and similarity to timescales observed in pentacene thin films lead us to conclude that this population loss is specific to well-ordered domains of the dendrimer[57–60]. However, the data also suggests that a portion of the triplet pair decay within the ensemble occurs in less ordered regions, where triplets are less mobile. For example, there exists an ensemble of triplet excitons that persist past this initial decay process, with a residual population of around 35% of the maximum.

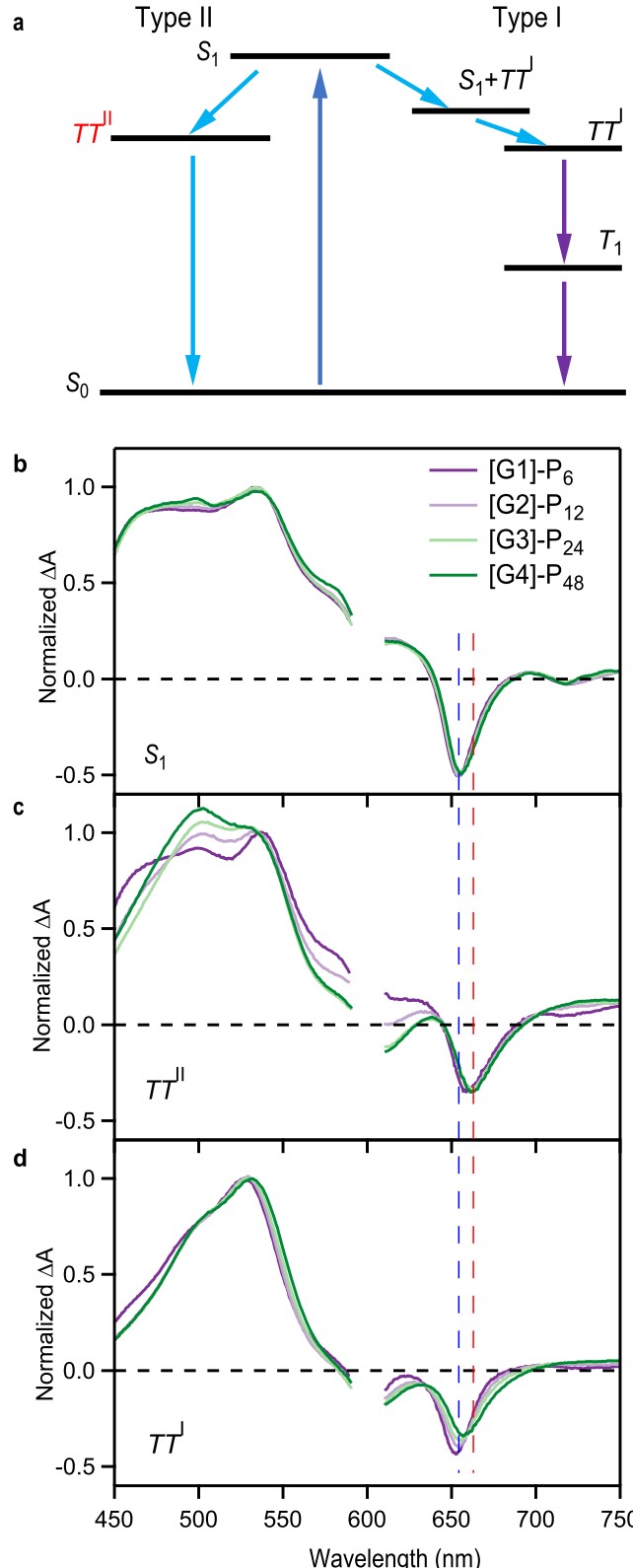

**Fig. 5 | Target analysis of transient absorption spectra based on the branched scheme. a** Branched kinetic scheme used for global and target analysis of femto- and nanosecond transient absorption data. Type-I species correspond to weakly interacting chromophore regions, while Type-II species correspond to more strongly interacting chromophores. Species-associated spectra derived from the model in (**a**) for **b** singlet excitons, **c** $TT^{II}$ triplet pairs, and (**d**) $TT^{I}$ triplet pairs. The blue dashed line is a guide to the eye that corresponds to the spectral position of the GSB for the singlet. The red dashed line corresponds to the GSB minimum for the $TT^{II}$ triplet pairs.

These triplets exhibit relatively long-lived decay kinetics (Fig. 4), with the average behavior captured in two generation-dependent, characteristic time constants spanning (100–250 ns) and (21–27 μs). The ~100 ns time constant is consistent with triplet–triplet annihilation processes observed in molecular dimers. The final ~20 μs lifetime is characteristic of individual, free triplet excitons rather than triplet pairs. A residual free triplet population has been previously identified in intramolecular SF materials as a characteristic of weaker inter-chromophore coupling[8,61].

To capture the full multiscale exciton dynamics of the SF dendrimers, we have constructed a model that partitions the spin conversion process into two distinct branches to account for the differing morphology of chromophores within a single dendrimer generation (Fig. 5a). This model is discussed in detail in the Supplementary Discussion. Briefly, we partition the dynamics into a fast triplet pair generation and decay segment associated with local regions of strong chromophore electronic coupling and a slower triplet pair generation and decay segment is associated with regions of weaker coupling. To be consistent with similar approaches used in SF nanoparticles, we adopt previously established naming conventions where Type I indicates weakly coupled chromophores with slower kinetics and Type II indicates strongly coupled chromophores with faster kinetics. The full broadband transient absorption spectra for all four dendrimer generations were modeled using this full model to obtain accurate time constants and recover characteristic transient spectra associated with intermediate species. From [G1]-$P_6$ to [G4]-$P_{48}$, the rates of formation and decay increase with the generation for both the short and long-lived triplet states, again suggesting strong electronic coupling on average in large generation dendrimers (summarized in Table 1). For the first generation [G1]-$P_6$, we observe the fast (8.6 ps) and slow (101.1 ps) triplet formation and the fast triplet decay (970 ps). We note that these values are almost identical to what is observed in nanoparticles of TIPS-pentacene with a similar value of $\varepsilon_{703}/\varepsilon_{655}$ (0.026) despite significant differences in the framework (covalent dendrimers versus non-covalent nanoparticle)[54].

More interestingly, the extracted transient spectra (Fig. 5b–d) show that the exciton dynamics reflect the inherent heterogeneity of a single macromolecule and that later generations show a larger degree of exciton migration within the macromolecule. At early times for which the singlet exciton is populated, the average position of the GSB corresponds to the steady-state absorption maximum (between 650 and 655 nm) (Fig. 5b) and is nearly identical for [G1]-$P_6$ to [G4]-$P_{48}$. As absorption is expected to be random among the chromophore sites in the dendrimer, these results suggest that the average local chromophore environment is similar for all generations. However, the GSB position of the Type II triplet pair state ($TT^{II}$) exhibits spectral shifts of ~8 nm relative to the singlet for all generations (Fig. 5c). The different GSB position $TT^{II}$ state means that they represent a small fraction of the total available sites within the dendrimer, even though ~60% of the population loss occurs at these sites. As the relative fraction of loss at these sites is only weakly dependent on generation (Fig. 4b), we conclude that $TT^{II}$ states primarily form due to pairwise chromophore interactions with preferential alignment for strongly bound triplet pairs. In contrast, the nature of Type I triplet pairs ($TT^{I}$) sites is generation-dependent, as is reflected in the variation in GSB minimum. For [G1]-$P_6$, the GSB position associated with $S_1$ and $TT^{I}$ are nearly identical, suggesting that the electronic environment of the absorbing site and the SF site are similar on average, i.e., minimum exciton migration occurs. However, in [G4]-$P_{48}$, we still observe a large red-shift of ~4 nm associated with $TT^{I}$ compared to $S_1$. This implies significant differences exist in the local chromophore coupling between the absorbing site and the SF site, i.e., exciton migration is occurring in later generations. This is consistent with numerical simulations showing a greater diversity of chromophore orientations in higher generations. We conclude that larger generations exhibit both a

larger quantity of strongly coupled chromophores and a greater diversity in chromophore–chromophore alignment compared to lower generations.

In summary, we designed and synthesized a series of TMP-core, bis-MPA dendrimers with TIPS-pentacene and TIPS-anthracene periphery in order to study the impact of multiexcitonic through-space interactions in dendritic architectures. The interactions between chromophores were investigated through SF dynamics and upconversion luminescence measurements. In both systems, efficient exciton migration is observed such that the multiexciton dynamics are dominated by a few hot spots that feature well-ordered chromophores. The number of hot spots increases as a function of generation. This behavior contrasts with macromolecular systems with more structural degrees of freedom (e.g., pendent polymers), where the dynamics are less dependent on the system size. Considering the number of studies of multiexciton processes in small molecules and bulk crystalline compounds, macromolecular systems provide new means to control multi-chromophore arrangements to modulate their coupling interactions. Macromolecular multiexcitonic materials are emerging as a platform to study mesoscale dynamics. This dimension reveals rich information that bridges our fundamental understanding between the small-molecule scale and bulk crystalline systems.

## Methods

### The synthesis of dendrimers

All commercially obtained reagents were used as received. Chemicals were purchased from Accela ChemBio, Alfa Aesar, Acros Organics, Oakwood Chemical, Sigma-Aldrich, and TCI America and were used as received without further purification. Anhydrous solvents were obtained from a Schlenk manifold with purification columns packed with activated alumina and a supported copper catalyst (Glass Contour, Irvine, CA). Unless stated otherwise, reactions were conducted in oven-dried glassware under an argon atmosphere. Hydroxyl-terminated dendrimers were synthesized according to the literature procedure.

$^1$H-NMR and $^{13}$C-NMR spectra were recorded on Bruker 300 MHz, 400 MHz, and 500 MHz (125 MHz for $^{13}$C) spectrometers. Data from the $^1$H-NMR and $^{13}$C-NMR spectroscopy are reported as chemical shifts (δ ppm) with the corresponding integration values. Coupling constants (J) are reported in hertz (Hz). Standard abbreviations indicating multiplicity were used as follows: s (singlet), d (doublet), t (triplet), q (quartet), and m (multiplet).

MALDI-TOF spectra were generated with a Bruker UltraFlex MALDI-TOF with a scout-MTP Ion source (Bruker Daltonics, Bremen, Germany), $N_2$-laser (337 nm), and a reflector. 2,5-Dihydroxybenzoic acid (DHB) or 2-[(2E)-3-(4-tert-Butylphenyl)-2-methylprop-2-enylidene] malononitrile (DCTB) were used as the matrix, sodium trifluoroacetic acid (NaTFA) as the ion, and THF as the solvent.

GPC were obtained on a Waters Alliance 2695 separation module equipped with a PL-aqua gel-OH 8-micron Mixed-M column (300 × 7.5 mm), a Waters 2998 Photodiode Array Detector, and a Waters 2414 Refractrometer Detector with an elution rate of 1 mL/min was used to run gel permeation chromatography (GPC). Stabilized THF was used as the eluent. Polystyrene standards were used to determine molecular weight and dispersity.

### Optical measurements

Absorption spectra were obtained on an Agilent Technologies Cary 60 UV–Vis Spectrophotometer.

Transient absorption measurements were carried out using a broadband pump–probe setup, which is pumped by a commercial Ti:Sapphire laser system (Coherent Astrella, 1 kHz) with an optical parametric amplifier. For femtosecond TA measurements, the supercontinuum probe light is generated by focusing the fundamental pulse (800 nm) into a sapphire plate. The probe light is split into signal and reference beams. The pump-probe delay was controlled by a mechanical delay line. For nanosecond TA measurements, the probe light is generated with a fiber laser (Leukos), and the delay times up to hundreds of microseconds are controlled by an electronic delay configuration. The pentacene was excited by 600 nm, which is consistent for fs- and ns-TA measurements. All pentacene dendrimer concentrations were kept below 50 μM in toluene.

Global and target analysis. The transient absorption spectra were further analyzed using the publicly available program Glotaran based on the statistical fitting package TIMP[62,63]. A branched target model and a sequential global model were performed for the fs and ns TA data, respectively.

Triplet photosensitization. The triplet sensitization experiments were performed on the same setup as the ns TA. A solution of ~10 mM anthracene in toluene, along with a low concentration of dendrimers, was excited by a 360 nm pulse. The triplet of dendrimers is generated by the collisional triplet energy transfer from the anthracene triplet, which is from the initial singlet through the intersystem crossing (ISC).

Upconversion experiments were performed on degassed chloroform solutions with the triplet sensitizer palladium (II) tetraphenyltetrabenzoporphyrin (PdTPTBP) at 50 μM. Dendrimer or TIPS-An-BE concentration was held low and constant for each set of experiments to minimize dendrimer intermolecular collisions while maintaining similar rates of sensitization. Due to the inherently low UC quantum yield ($\Phi_{UC}$) at concentrations measured, we have chosen to directly compare the UCPL for each set of samples. The anthracene dendrimer solutions were excited with a 635 nm laser, and their UCPL was recorded. All measurements were completed behind a 600 nm short-pass filter to remove scattered excitation light. Integrated UCPL intensity values were calculated by integrating the area under the UCPL intensity spectrum between 400 and 600 nm, and standard deviation was determined from triplicate measurements at each concentration.

## Data availability

The steady-state spectra of pentacene dendrimer and upconversion data of anthracene dendrimer for Fig. 2 and the raw transient absorption data used to generate Figs. 3 and 4 will be available via the Dryad public repository: https://doi.org/10.5061/dryad.7m0cfxq0w. Additional data are available upon request.

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

## Acknowledgements

This work was supported by the National Science Foundation under grants DMR-2004683, DMR-2003444 A.C. and DMR-2004678. E.M.C. thanks the NSF GRFP (grant No. 1644869). K.R.P. thanks the Department of Defense for a National Defense Science and Engineering (NDSEG) Fellowship. P.N. acknowledges the support of a Stanford Graduate Fellowship in Science & Engineering (SGF) as a Gabilan Fellow. F.N. and M.M. acknowledge Knut and Alice Wallenberg Foundation, Grant KAW (2018.0452-WWSC 2.0, 2019.0002 and 2017.0300). This research used the Theory and Computation facility of the Center for Functional Nanomaterials (CFN), which is a U.S. Department of Energy Office of Science User Facility at Brookhaven National Laboratory under Contract No. DE-SC0012704.

## Author contributions

M.Y.S. and L.M.C. oversaw the project. E.M.C., K.R.P., J.Z. synthesized and characterized the molecules. The upconversion photoluminescence measurements and characterization were carried out by E.M.C., P.N., and D.N.C. Dendrimer synthesis and characterization were carried out by F.N. and M.M. G.H. collected the transient absorption spectroscopy and G.H. and M.Y.S. carried out data analysis. Simulations were carried out by A.C. The paper was written by G.H., E.M.C., M.Y.S., and L.M.C., with contributions from all authors.

## Competing interests

D.N.C. is cofounder of and Chief Scientific Advisor to Quadratic 3D, Inc. The remaining authors declare no competing interests.
