## [Peer Review File · Nature Communications]

Promoting multiexciton interactions in singlet fission and triplet fusion upconversion dendrimersREVIEWER COMMENTS

Reviewer #1 (Remarks to the Author):

In this paper, the authors synthesize four generations of dendrimers of tri-isopropylethynyl(TIPS) substituted pentacene and anthracene to examine how the high density of chromophores tethered together would affect singlet fission or triplet-triplet annihilation (TTA). The first generation has 6 acenes. The number of acenes per dendrimer increases till the fourth generation with 48 acenes. These are beautiful macromolecules purified by GPC, characterized by MALDI, steady-state photoluminescence and transient absorption (TA) spectroscopy. Unfortunately, the 3rd and 4th generation anthracene dendrimers suffer from excimer formation, resulting in a low upconversion quantum yield that is swept under the carpet. Considering the fact that Bossanyi and Clark have shown a high spin-statistical factor, f , for rubrene resulting from molecular orbital overlap and the energetics of the T2 state vs S1 state that allows excitons to be recycled, it would be interesting to see if diphenylanthracene or rubrene dendrimers would show enhanced f values and better photon upconversion. The linear alkyne in the TIPS-aceytene group unfortunately results in excimer formation.

Ultrafast TA shows that the pentacene dendrimers house the pentacene molecules in different environments akin to disordered pentacene nanoparticles. Thus some pentacene molecules are in the position to perform the usual fast singlet fission, while some are slow. Though it is challenging to measure the relative orientation of each chromophore in a dendrimer, the authors have done a commendable job with some simulations.

Overall, these dendrimers are an intriguing concept and the experiments here are interesting and conducted well. The paper is also well-written.

Reviewer #2 (Remarks to the Author):

This paper uses dendrimers to control the local concentration of pentacene chromophores for singlet fission studies and anthracene chromophores for upconversion studies. Each of these photophysical processes are detailed and are used as indicators of the advantages of

using dendrimers for these purposes. The authors argue in terms of the relative geometries of the chromophores, but there is no substantive structural information. The increase in local concentration of peripheral groups in higher generation dendrimers is also a well-established phenomenon. To that end, the results presented in this paper do not offer any significant new insights. Some of the absorbance ratios offered as structural evidence are relatively small and may result from other causes. The fluorescence quenching resulting from singlet fission increases as the generation number increases, yet again, there is really no specific structural information that can be gleaned from the data. Despite these criticisms, the authors have done careful photophysical work on these systems. The systems themselves are problematic with regard to establishing firm conclusions based on the data.

Reviewer #3 (Remarks to the Author):

The manuscript by He and coworkers reports dendrimers bearing acene chromophores which are either prototypical singlet fission or triplet-triplet annihilation materials. As the crowding of chromophores increases with the number of generations, owing to the exponential increase in the number of chromophores while the surface area grows with the square, singlet fission is seen to proceed more rapidly, and the generated triplets are shorter-lived. TTA-UC is seen to generate excimeric emission where the the chromophores are most crowded. This would usually be considered detrimental.

One key point of interest that is not fully explained is why the dendrimers outperform solution-phase TIPS-anthracene in terms of upconversion. The comparison is based on chromophore concentration in Figure 2. One cannot judge the advantage of the dendrimer without certain photophysical measurements being performed. At 1 μM emitter concentration, we expect very poor triplet harvesting from of the porphyrin. To judge the effectiveness of the triplet harvesting, one needs the phosphorescence lifetimes with and without the quencher. We also need to know the pump intensity and therefore the excitation rate of the porphyrin. There needs to be at least 2 excited chromophores in the one dendrimer for TTA-UC to proceed! Do we expect excited dendrimers to collide? As it stands, it is not possible to judge the reason for the increased efficacy of the dendrimer over the equivalent concentration of free chromophore. The stated explanation is without basis.

Some other things:

Page 2. "lowest energy excited state" is nearly always T1. Wording needs revision.

Page 2. The reference list for SF solar cells should include MacQueen's tetracene on silicon cell (Materials Horizons 2018), and Einzinger's report in Nature.

Page 10. The observation of a triplet is not proof of singlet fission. ISC could be occurring.

We thank the reviewers for their assessment of our work and feedback to improve the dissemination of our research findings.

REVIEWER COMMENTS

Reviewer #1 (Remarks to the Author):

In this paper, the authors synthesize four generations of dendrimers of tri-isopropylethynyl (TIPS) substituted pentacene and anthracene to examine how the high density of chromophores tethered together would affect singlet fission or triplet-triplet annihilation (TTA). The first generation has 6 acenes. The number of acenes per dendrimer increases till the fourth generation with 48 acenes. These are beautiful macromolecules purified by GPC, characterized by MALDI, steady-state photoluminescence and transient absorption (TA) spectroscopy.

We appreciate the strong words of support from the reviewer.

Unfortunately, the 3rd and 4th generation anthracene dendrimers suffer from excimer formation, resulting in a low upconversion quantum yield that is swept under the carpet. Considering the fact that Bossanyi and Clark have shown a high spin-statistical factor, f , for rubrene resulting from molecular orbital overlap and the energetics of the T2 state vs S1 state that allows excitons to be recycled, it would be interesting to see if diphenylanthracene or rubrene dendrimers would show enhanced f values and better photon upconversion. The linear alkyne in the TIPS-aceytene group unfortunately results in excimer formation.

We thank the reviewer for this comment, but we respectfully disagree with the premise that excimer formation in higher generations is “unfortunate,” leading to low UCPL yields. In Figure 3d (see below), the total integrated UCPL yield continues to increase, even after excimer formation is observed (Figure 3c). Furthermore, the formation of excimers in higher generations has not been “swept under the carpet.” In fact, excimer formation is a key supporting piece of data for the main message of this manuscript, which is that in *higher dendrimer generations, a few highly ordered sites dominate the multiexciton dynamics.*

Figures 3 c & d. (See caption in the main text).

The unique characteristics of our dendritic macromolecules allows us to identify this apparent anomaly and explore important unresolved questions in multiexciton systems including the impact

of excimer formation in these “hot spots” on the performance of the triplet-triplet upconversion process (and in singlet fission). We found that there are mixed conclusions: If only the total integrated quantum yield is considered, then high generation dendrimers in dilute solution enable both long penetration depths and efficient UCPL (Figure 3d). Monomers in solutions with the same optical density (effective concentration) exhibit much lower yields. However, if the key metric of interest is the apparent anti-Stokes shift, then higher order anthracene dendrimers will yield a lower value, as compared to the monomer.

We agree that other chromophores would be of interest to see how the dendrimer architecture impacts their multiexciton dynamics. Importantly, this study provides a framework that can be used for these types of follow up studies that are of interest to a wide community. As the reviewer suggests, these systems can be likely be optimized to exhibit both high yields and high apparent Stokes shifts. However, this is not the goal of our manuscript, which is primarily concerned with understanding how the dendrimer motif affects local chromophore coupling.

We have edited the abstract and main text to reinforce this primary messaging. The abstract now reads:

“Singlet fission (SF) and triplet-triplet annihilation upconversion (TTA-UC) are two multiexciton processes intimately related to the dynamic interaction between one high-lying energy singlet and two low-lying energy triplet excitons. Here, we introduce a series of dendritic macromolecules that serve as platform to study the effect of interchromophore interactions on the dynamics of multiexciton generation and decay as a function of dendrimer generation. The dendrimers (generations 1-4) consist of trimethylolpropane (TMP) core and 2,2-bis(methylol)propionic acid (bis-MPA) dendrons that provide exponential growth of the branches, leading to a corona decorated with pentacenes for SF or anthracenes for TTA-UC. The findings reveal a trend where a few highly ordered sites emerge as the dendrimer generation grows, dominating the multiexciton dynamics, as deduced from optical spectra, and transient absorption spectroscopy. While the dendritic structures enhance TTA-UC at low annihilator concentrations in the largest dendrimers, the paired chromophore interactions induce a broadened and red-shifted excimer emission. In SF dendrimers of higher generations, the triplet dynamics become increasingly dominated by pairwise sites exhibiting strong coupling (Type II), which can be readily distinguished from sites with weaker coupling (Type I) by their spectral dynamics and decay kinetics”

Additionally, the discussion on pages 2-4 had significant modifications.

Ultrafast TA shows that the pentacene dendrimers house the pentacene molecules in different environments akin to disordered pentacene nanoparticles. Thus some pentacene molecules are in the position to perform the usual fast singlet fission, while some are slow. Though it is challenging to measure the relative orientation of each chromophore in a dendrimer, the authors have done a commendable job with some simulations.

Overall, these dendrimers are an intriguing concept and the experiments here are interesting and conducted well. The paper is also well-written.

We appreciate the fortifying comments and would like to emphasize that our analysis of the singlet fission dynamics goes hand-in-hand with our analysis of the UCPL process. In both cases, the data

supports the notion that higher generations facilitate the formation of a few “hot spots” with strong chromophore-chromophore coupling and that these minority sites dominate the multi-exciton dynamics.

Reviewer #2 (Remarks to the Author):

This paper uses dendrimers to control the local concentration of pentacene chromophores for singlet fission studies and anthracene chromophores for upconversion studies. Each of these photophysical processes are detailed and are used as indicators of the advantages of using dendrimers for these purposes.

The authors argue in terms of the relative geometries of the chromophores, but there is no substantive structural information.

We thank the reviewer for the assessment of our manuscript, but we respectfully disagree with the premise that there is no “substantive structural information” on our dendrimer system. We have provided extensive characterization of the dendrimers by NMR, size exclusion chromatography, mass spectrometry, optical spectroscopy, and molecular dynamics simulations. These data provide information of the well-defined macromolecular structures ($D \sim 1.02$), which indicate that pairwise chromophore interactions dominate in all generations, although the degree of interchromophore coupling is heterogeneous. Furthermore, the probability of having a well-ordered pair is higher in higher generations. We note that these materials are soft and dynamic at room temperature, with significant rotational degrees of freedom. As such, we maintain that we provided exhaustive structural information.

The increase in local concentration of peripheral groups in higher generation dendrimers is also a well-established phenomenon. To that end, the results presented in this paper do not offer any significant new insights.

We agree with the reviewer as to their point that the increase in concentration in peripheral groups as a function of generation is established, but that is not the thesis of this work. Extensive literature on dendrimers provides numerous examples of the diverse dynamic behavior of dendrimers, which depend on the identity of the core, dendrons, generation, corona, etc.[10.1021/cr900327d] We used a bis-MPA based dendrimer to begin our studies of exciton dynamics in macromolecular systems, and it is important to note that there are various ways to modify the peripheral coupling interactions. Many architectures exist that provide rigidity or flexibility to the corona functional groups, such as bowtie dendrimers, dendritic polymers, and many other configurations that can lead to tunable local interactions of the peripheral groups. We consider the data reported here to be a first-of-its-kind study focused on establishing a foundation for probing the multiexciton dynamics as a function of generation, including the impact of *local pairwise interactions*. We are not focused on modifying the flexibility of the dendrons/corona. This work is unique and important because multiexciton processes—being inherently multichromophore systems—are sensitive reporters of changes in intimate chromophore-chromophore interactions. In fact, the conclusion is that on average, the structure of the pairwise interactions of the acenes on the dendrimer is only weakly affected as a function of generation. Moreover, the development of a few “hot spots” can be uniquely identified using multiexcitonic chromophores. These hot spots almost completely dominate the multiexciton photophysics.

As noted in the response to Reviewer 1, we made significant changes to the abstract and introductory discussion.

Some of the absorbance ratios offered as structural evidence are relatively small and may result from other causes. The fluorescence quenching resulting from singlet fission increases as the generation number increases, yet again, there is really no specific structural information that can be gleaned from the data. Despite these criticisms, the authors have done careful photophysical work on these systems. The systems themselves are problematic with regard to establishing firm conclusions based on the data.

Changes in the absorbance ratios are significant but represent only a single piece of supporting data for the conclusions. We note that the changes in absorbance ratios agree well with previous work on acene aggregates and that they are completely consistent with our conclusions from time-resolved studies. To support our methodology, we cite prior art on pentacene tetramer films and particles to show how the small red-shift of the absorbance is a signal of weak coupling, which is beneficial to the long-lived triplet. (10.1016/j.chempr.2019.06.007; 10.1039/C7MH00303J; 10.1021/acs.jpcc.5b11353). Importantly, we do not observe crystalline motifs that are analogous to the condensed phase morphology, in which dramatic red-shifts are observed, but instead observe heterogeneous strong and weak coupling regions similar to acene nanoparticles.

Reviewer #3 (Remarks to the Author):

The manuscript by He and coworkers reports dendrimers bearing acene chromophores which are either prototypical singlet fission or triplet-triplet annihilation materials. As the crowding of chromophores increases with the number of generations, owing to the exponential increase in the number of chromophores while the surface area grows with the square, singlet fission is seen to proceed more rapidly, and the generated triplets are shorter-lived.

TTA-UC is seen to generate excimeric emission where the the chromophores are most crowded. This would usually be considered detrimental.

We thank the reviewer for the assessment of our manuscript. We would argue that the question of whether aggregation and excimer formation are detrimental is an ambiguous question. “Good” interchromophore interactions are essential to formation of triplet pairs that are needed to facilitate the upconversion process, but “bad” ones can be parasitic. To our knowledge, a detailed understanding of what constitutes “good” and “bad” has not been clearly articulated. In fact, our study uniquely provides key insights into this issue.

The dendrimers largely exhibit weak interchromophore coupling, but stronger coupled “hot spots” emerge in higher generations. These hot spots have important consequences for the triplet-triplet upconversion process, with mixed results. If only the total integrated quantum yield is considered, then high generation dendrimers in dilute solution enable both long penetration depths and efficient UCPL. Monomers in solutions with the same optical density (effective concentration) exhibit much lower yields.

However, if the key metric of interest is the apparent anti-Stokes shift, then higher order anthracene dendrimers will yield relatively smaller shifts compared to the monomers. As Reviewer 1 suggests,

these systems can likely be optimized to exhibit both high yields and high apparent anti-Stokes shifts. However, efficiency is not the goal of our manuscript. We are primarily concerned with understanding how the dendrimer motif affects local chromophore couplings in order to build on this foundation to engineer advanced systems to make them more efficient.

We have included a more explicit discussion of these factors in the main manuscript (pages 3-4).

One key point of interest that is not fully explained is why the dendrimers outperform solution-phase TIPS-anthracene in terms of upconversion. The comparison is based on chromophore concentration in Figure 2. One cannot judge the advantage of the dendrimer without certain photophysical measurements being performed. At 1 μM emitter concentration, we expect very poor triplet harvesting from the porphyrin.

The dendrimer structure here is used as a platform for detailed investigations of the multichromophore dynamics in macromolecular architectures. Insights gained in these studies, that a few key sites dominate the UC process, will be critical to the design of future macromolecular architectures with optimized multiexcitonic processes. In the systems discussed here, our goals are fundamental studies, and we have clarified any discussion that might imply we have outperformed monomer-based systems. While it is notable that the dendrimer architecture enables more efficient UCPL in the regime of very low concentrations and long penetration depths, we recognize that these are not ideal device conditions. Still, these results reflect the fact that the dendrimer motif serves to organize the chromophores and increase their *local* concentration so the probability of triplet-triplet annihilation is increased. The choice of the low (1 μM) concentrations used here are ideal conditions to study the multielectron dynamics, even though they result in low overall UCPL yields.

To address the concerns of the reviewer, we have added additional data to the SI that shows the UC yield over a broad range of annihilator concentrations from 1 - 50 μM , which corresponds to an effective monomer concentration of $\sim 300 \mu\text{M}$ ([G1]-A₆). We find that generation 2 gives the highest overall UCPL yields, comparable (but not exceeding) that of the monomer. We note that this data provides a better understanding of the factors that could be used to take advantage of dendritic architectures, in which “hot spots” can be designed with optimal coupling.

To judge the effectiveness of the triplet harvesting, one needs the phosphorescence lifetimes with and without the quencher. We also need to know the pump intensity and therefore the excitation rate of the porphyrin. There needs to be at least 2 excited chromophores in the one dendrimer for TTA-UC to proceed! Do we expect excited dendrimers to collide? As it stands, it is not possible to judge the reason for the increased efficacy of the dendrimer over the equivalent concentration of free chromophore. The stated explanation is without basis.

We emphasize that inter-dendrimer collisions are not necessary to achieve UCPL. Instead, we require multiple collisions between a sensitizer and a specific dendrimer with a collision rate that exceeds the triplet lifetime. This collision rate can be readily estimated. For a bimolecular reaction induced by collisions with stick boundary conditions, the diffusion-limited association rate k_D is estimated from the Stokes-Einstein relationship as $k_D = 8RT/3\mu$, where R is the ideal gas constant, T is temperature, and μ is the solvent viscosity. Under the conditions used in this study,

the absorbance of the sensitizer is ~ 2 in chloroform (with viscosity $\mu_{chl} = 0.54$ mPa s) and the irradiance is 43 W/cm², we find that $k_D = 1.2 \times 10^{11} M^{-1}s^{-1}$. Therefore, the mean collision time ($\tau_D = 1/k_D[D]$) is estimated to be $\tau_D = 1.6$ μ s with an effective photoexcited sensitizer concentration of 5 μ M ($\sim 10\%$ of the total concentration). As such, we expect multiple triplet transfer events to occur to a single dendrimer on time scales shorter than the triplet lifetime of TIPS-anthracene (~ 35 us).

These estimates can be verified using transient absorption spectroscopy to measure the triplet sensitization yield under various conditions. At total concentrations above 5 μ M of annihilator (well below the estimate above), we start to measure a significant triplet quenching in the sensitizer. At ~ 50 μ M of sensitizer, we estimate that the triplet transfer yield is $\phi = \frac{k_{[c]=50\mu M}}{k_{[c]=50\mu M} + k_0} = \frac{0.034}{0.034 + 0.05} = 0.4$. Using Poisson statistics, we estimate that the probability that an individual dendrimer undergoes two triplet transfer events is $\sim 5\%$. These data are consistent with the idea that significant UCPL can be observed under the conditions reported here.

These data have been added to the Supplementary information. (section II “Triplet-triplet Upconversion Photoluminescence”. pages 4-7).

Some other things:

Page 2. "lowest energy excited state" is nearly always T1. Wording needs revision.

We thank the reviewer for pointing this out. This sentence has been fixed as following:

“When the energy of biexciton is less than that of the lowest singlet state”.

Page 2. The reference list for SF solar cells should include MacQueen's tetracene on silicon cell (Materials Horizons 2018), and Einzinger's report in Nature.

The citations have been added.

Page 10. The observation of a triplet is not proof of singlet fission. ISC could be occurring.

For pentacene dendrimers, we observed fast singlet decay with the corresponding triplet rise in less than 100 ps with high triplet yield, which is different from the ISC process. In addition, we observed the triplet-triplet annihilation process from the ns-TA data.

REVIEWERS' COMMENTS

Reviewer #1 (Remarks to the Author):

The authors have responded adequately to all reviewer comments, especially pointing out that at low concentrations, the excimeric emission from the dendrimers can exceed the emission from the small molecule precursor, TIPS-Anthracene. This is unexpected and interesting, a valuable addition to the field. It is exceedingly difficult to characterize interchromophore interactions in these dendrimers during singlet fission/ upconversion.

Reviewer #2 (Remarks to the Author):

I stand by my original comments that the fluxional nature of the dendrimers results in a distribution of structural motifs that are difficult to relate to the photophysics. I leave it to the editor to decide whether the authors have made a convincing case to the contrary.

Reviewer #4 (Remarks to the Author):

The paper by He et al presents beautiful dendrimer systems and their applicability for singlet fission in the case of pentacenes and photon upconversion in the case of anthracenes. The paper is heavier on the singlet fission part which makes it somewhat unbalanced, this reviewer would have wished the same detailed study of the upconversion process. That said, adding those experiments would mean additions of a significant amount of work.

The authors argue well for their conclusions and the data presented support their conclusions although this reviewer believes that there are still some question marks standing. That said, I believe this paper would add a significant contribution and is worth publishing in its current form.

We thank the reviewers for their assessment of our work and feedback to improve the dissemination of our research findings.

REVIEWER COMMENTS

Reviewer #1 (Remarks to the Author):

The authors have responded adequately to all reviewer comments, especially pointing out that at low concentrations, the excimeric emission from the dendrimers can exceed the emission from the small molecule precursor, TIPS-Anthracene. This is unexpected and interesting, a valuable addition to the field. It is exceedingly difficult to characterize interchromophore interactions in these dendrimers during singlet fission/ upconversion.

We are delighted to hear that the manuscript reads clearly in the revised version, and we thank the reviewer for their time and effort reviewing the manuscript.

Reviewer #2 (Remarks to the Author):

I stand by my original comments that the fluxional nature of the dendrimers results in a distribution of structural motifs that are difficult to relate to the photophysics. I leave it to the editor to decide whether the authors have made a convincing case to the contrary.

We thank the reviewer for the additional feedback, and we maintain that we have demonstrated strong correlations between the structure of the dendrimer and the resulting structure in two distinct systems exhibiting singlet fission and upconversion respectively. Our results are supported by extensive characterization of the dendrimers by NMR, size exclusion chromatography, mass spectrometry, optical spectroscopy, and molecular dynamics simulations.

Reviewer #4 (Remarks to the Author):

The paper by He et al presents beautiful dendrimer systems and their applicability for singlet fission in the case of pentacenes and photon upconversion in the case of anthracenes. The paper is heavier on the singlet fission part which makes it somewhat unbalanced, this reviewer would have wished the same detailed study of the upconversion process. That said, adding those experiments would mean additions of a significant amount of work.

We thank the reviewer for the assessment of our manuscript.

The authors argue well for their conclusions and the data presented support their conclusions although this reviewer believes that there are still some question marks standing. That said, I believe this paper would add a significant contribution and is worth publishing in its current form.

We appreciate the reviewer's comments in support of publication. We agree that a number of studies are possible based on this new concept, looking at chromophore-chromophore interactions, and the goal of this study is to stimulate other research that can unveil new insights.